# Three-Dimensional Printing of Curcumin-Loaded Biodegradable and Flexible Scaffold for Intracranial Therapy of Glioblastoma Multiforme

**DOI:** 10.3390/pharmaceutics13040471

**Published:** 2021-03-31

**Authors:** Ruixiu Li, Yunmei Song, Paris Fouladian, Mohammad Arafat, Rosa Chung, Jarrod Kohlhagen, Sanjay Garg

**Affiliations:** 1Pharmaceutical Innovation and Development (PIDG) Group, Clinical and Health Sciences, University of South Australia, Adelaide 5000, Australia; liyry032@mymail.unisa.edu.au (R.L.); may.song@unisa.edu.au (Y.S.); paris.fouladian@mymail.unisa.edu.au (P.F.); mohammad.arafat@mymail.unisa.edu.au (M.A.); rosa.chung@unisa.edu.au (R.C.); 2Applied Chemistry and Translational Biomaterials (ACTB) Group, Clinical and Health Sciences, University of South Australia, Adelaide 5000, Australia; kohjs002@mymail.unisa.edu.au

**Keywords:** three-dimensional printing, curcumin, polycaprolactone, controlled release, glioblastoma multiform

## Abstract

A novel drug delivery system preventing Glioblastoma multiforme (GBM) recurrence after resection surgery is imperatively required to overcome the mechanical limitation of the current local drug delivery system and to offer personalised treatment options for GBM patients. In this study, 3D printed biodegradable flexible porous scaffolds were developed via Fused Deposition Modelling (FDM) three-dimensional (3D) printing technology for the local delivery of curcumin. The flexible porous scaffolds were 3D printed with various geometries containing 1, 3, 5, and 7% (*w*/*w*) of curcumin, respectively, using curcumin-loaded polycaprolactone (PCL) filaments. The scaffolds were characterised by a series of characterisation studies and in vitro studies were also performed including drug release study, scaffold degradation study, and cytotoxicity study. The curcumin-loaded PCL scaffolds displayed versatile spatiotemporal characteristics. The polymeric scaffolds obtained great mechanical flexibility with a low tensile modulus of less than 2 MPa, and 4 to 7-fold ultimate tensile strain, which can avoid the mechanical mismatch problem of commercially available GLIADEL wafer with a further improvement in surgical margin coverage. In vitro release profiles have demonstrated the sustained release patterns of curcumin with adjustable release amounts and durations up to 77 h. MTT study has demonstrated the great cytotoxic effect of curcumin-loaded scaffolds against the U87 human GBM cell line. Therefore, 3D printed curcumin-loaded scaffold has great promise to provide better GBM treatment options with its mechanical flexibility and customisability to match individual needs, preventing post-surgery GBM recurrence and eventually prolonging the life expectancy of GBM patients.

## 1. Introduction

Glioblastoma multiforme (GBM) is the most aggressive and lethal form of malignant brain cancer, which, according to the World Health Organisation (WHO), is classified as a Grade IV brain tumour [1,2]. The infiltrative nature of GBM makes it exceedingly hard to treat and invariably fatal [3]. Despite the optimal treatment of surgical resection followed by radiotherapy and chemotherapy, the five-year survival rate of GBM patients is still less than 10% [2,4,5]. Seventy percent of GBM recurrence happens within the surgical margin [6]. The GLIADEL wafer is the most widely used biodegradable polymer-based implant for GBM treatment [7]. During neurosurgery, GLIADEL wafers are inserted and attached to the resection margin to release Carmustine (BCNU) locally for a week to eliminate residual cancer cells. This kind of local polymeric drug delivery system for implantation during surgical removal of tumours not only circumvent the blood-brain barrier but also minimise undesired systemic exposure [4,8]. However, a few drawbacks prevent GLIADEL wafers from fulfilling clinical expectations. For instance, the solid wafers suffer from a mechanical mismatch with brain tissue, which incurs risks of neuroinflammatory and neurodegenerative response [9,10,11]. In addition, the solid wafers cannot fully cover the irregular cavity wall and might be easily detached [4]. Therefore, the limitations of GLIADEL wafer and the clinical demands for higher efficacy and low side-effects led us to explore a better alternative treatment option against GBM.

Three-dimensional (3D) printing technology is a revolutionary breakthrough in the era of personalised medicine [12]. 3D printing brings numerous advantages to fulfil personalised treatment, especially for the manufacture of pharmaceutical implants. Implantable delivery systems can be fabricated with the desired shape and structures to fit the tumour-resected site. The porosity and surface area can be easily altered by computer-aided design (CAD) to achieve predictable release kinetic [13,14,15]. Besides, its rapid prototyping nature makes it possible to real-timely print customised implants for individual patients at a low cost [15]. Here we utilise Fused Deposition Modelling (FDM) technology for the fabrication of anti-GBM polymeric scaffold, which is one of the most investigated 3D printing technologies and a well-established manufacturing tool to fabricate polymer-based implants. Polycaprolactone (PCL) has been extensively used as the drug loading matrix for implantable devices in different applications [16,17,18]. It is an FDA-approved polymer with great biodegradability, biocompatibility, and flexibility. Here we choose PCL as the scaffold matrix not only because it can be used safely but also provide the proposed scaffolds with great drug loading capacity and flexibility to overcome the mechanical limits of GLIADEL wafer [19].

Naturally derived complementary and nutraceutical compounds have been extensively studied for their beneficial therapeutic effects and low toxicity profiles. Curcumin is found in the rhizome of turmeric (Curcuma longa), which is a traditional medicinal herb extensively used for centuries in South Asia [20]. Curcumin has shown great promise in clinical oncology, due to its varied properties such as chemopreventive, chemotherapeutic, radiosensiblising, and chemosensiblising activities against a wide range of malignant cancers [21,22,23]. The great potential of curcumin against glioblastomas (GBMs) has been well studied both in vitro and in vivo [24,25]. Curcumin has been proved to have pharmacological effects against cancer stem cells, which is the major cause of therapy resistance of glioblastoma [26]. As a nutraceutical compound, curcumin has been proved in clinical data that it can be safely used in human at a high dose with low toxicity [25]. On the resection margin, both unresectable cancer cells and a large number of healthy brain cells are exposed. The local high concentration of curcumin might help to raise curative effects against cancer cells with limited toxicity on healthy cells. Besides, curcumin also benefits neuron regeneration after surgery [27]. Curcumin has been proved with at least 10 known in vitro neuroprotective actions [27]. Because of the beneficial anticancer effects of curcumin, here we chose it as the model drug. The main drawbacks that prevent curcumin from its clinical use are its poor water solubility, poor absorption, and fast metabolism in plasma, with a short elimination half-life of 8.64 ± 2.31 (i.v.) minutes [20,28,29]. Compared to oral administration or intravenous injection, local delivery of curcumin via PCL scaffold can circumvent BBB permeation and gastrointestinal absorption, allowing more drug reaching and staying in the target site for a prolonged period. Therefore, the local polymeric implant could be an ideal carrier for the efficient delivery of curcumin for potential therapeutic approaches into GBM.

In this study, we aim to fabricate biodegradable flexible scaffolds, incorporated with curcumin using FDM 3D printing technology. The versatile geometries of scaffolds were achieved with CAD and different drug loading doses scaffolds were prepared. A series of characterisation studies and in vitro evaluations demonstrated the achievements of the objectives of this project.

## 2. Materials and Methods

### 2.1. Materials

Curcumin (>97%) was obtained from Chem-Supply (Port Adelaide, SA, Australia). Polycaprolactone (PCL, Mn 80,000) was a gift from Guangdong Jin Shengnan New Material Technology (Guangzhou, Guangdong, China). Acetonitrile (HPLC grade) and Dimethyl sulfoxide (DMSO) was purchased from Merck (Bayswater, VIC, Australia). Tetrahydrofuran (THF) and citric acid were obtained from Chem-Supply (Port Adelaide, SA, Australia). MTT (3-(4,5-dimethylthiazol-2-yl)-2,5-diphenyltetrazolium bromide) was purchased from Life Technologies Australia (Mulgrave, VA, Australia). Minimum Essential Medium Eagle (MEM), Fetal Bovine Serum (FBS), L-glutamine, Dulbecco’s Phosphate Buffered Saline (PBS), and sodium pyruvate solution were purchased from Sigma-Aldrich (Castle Hill, NSW, Australia).

### 2.2. Preparation of Curcumin-Loaded PCL Films via Solvent Casting

To guarantee the homogeneous dispersion of curcumin in PCL after hot-melt extrusion, curcumin-loaded PCL films as well as a blank film were prepared in advance using the solvent casting method. To prepare different loading groups, PCL pellets and curcumin were accurately weighed and dissolved with Tetrahydrofuran (THF) in a sealed glass bottle (Table 1). The solutions were heated in a water bath at 55 °C, 6 h to form homogeneous 20% *w*/*v* solutions. The solutions then were cast into Petri dishes in a dark fume hood heated at 40 °C for 2 days to evaporate the solvent. Curcumin-loaded films were prepared with four loadings of 1, 3, 5, and 7% *w*/*w*. The same method was used to prepare the blank films with no drug added. The films were manually shredded into small pieces (<5 × 5 mm) for Hot-Melt extrusion.

### 2.3. Hot-Melt Extrusion (HME) of Curcumin-Loaded PCL Filaments

The blank and curcumin-loaded PCL filaments were hot-melt extruded with polymeric film pieces using a Composer 350 filament extruder (3devo B.V., Utrecht, The Netherlands). The extruding temperatures of each heating zones of the extruder from zone 1 to 4 were 70, 75, 80, and 95 °C, respectively and the extrusion speed was 4.5 rpm. The diameters of the extruded filaments were measured automatically by the filament extruder. Only filaments within the diameter range of 2.85 ± 0.15 mm were adopted for FDM printing.

### 2.4. Fused Deposition Modelling (FDM) 3D Printing of Curcumin-Loaded Scaffolds

The Computer-aided Design (CAD) of scaffold models with specific dimension (60 × 60 mm), thicknesses (0.1, 0.2, and 0.4 mm), and pore shapes types (honeycomb, square 1, square 2, triangle) were performed using Solidworks (Education Edition 2018, Dassault Systemes, Vélizy-Villacoublay, France) software. The inner-circle diameters of different pore shape type honeycomb, square 1, square 2, triangle were 2.0, 1.2, 0.8, and 2.0 mm, respectively. The interval between each pore was 0.8 mm (the minimum value to ensure printability). The value of surface area and volume of each model was generated automatically by Solidworks.

The resulting stereolithography files were sliced by Ultimaker Cura^®^ software (Ultimaker B.V., Utrecht, The Netherlands) with 3D printing parameters as shown in Table 2. The resulting Cura files were read by the Ultimaker S5 FDM 3D printer (Ultimaker B.V., Utrecht, The Netherlands). After feeding the filaments to the printing nozzle (diameter 0.4 mm), honeycomb scaffolds with different loadings (blank, 1%, 3%, 5% and 7%, *w*/*w*) were 3D printed with a thickness of 0.2 mm. While the thickness remained 0.2 mm, 7.0% *w*/*w* scaffolds were printed with different types of pore shapes (honeycomb, square 1, square 2, triangle). The honeycomb scaffolds (7.0%, *w*/*w*) were printed with 1, 2, and 4 layers to generate scaffolds with different thicknesses (0.1, 0.2, and 0.4 mm). The mass of different loading scaffolds (1%, 3%, 5% and 7%, *n* = 5) were weighed.

### 2.5. Scanning Electron Microscopy (SEM)

The surface morphology of pure curcumin powder, the blank, and curcumin-loaded (7% *w*/*w*) scaffolds was studied with a MERLIN™ Scanning Electron Microscope (Jena, Thuringia, Germany). The samples were coated with platinum by sputter-coating before analysis with an automatic sputter coater (Agar Scientific Ltd., Stansted, Essex, UK). An accelerating voltage of 2 kV was used to obtain SEM images.

### 2.6. Photoacoustic Fourier-Transform Infrared Spectroscopy (Pa-FTIR)

Fourier-transform infrared (FTIR) spectroscopy of pure curcumin powder, blank, and curcumin-loaded scaffold was performed on Thermo Nicolet Magna 750 FTIR spectrometer (Thermo Nicolet, WI, USA) with 256 scans and a mirror velocity of 0.158 cm/s under helium purge. The spectral range was from 400 to 4000 cm^−1^.

### 2.7. X-ray Diffraction (XRD)

XRD analysis of pure curcumin powder, blank, and curcumin-loaded PCL scaffolds (7% *w*/*w*) was performed using Empyrean XRD Diffractometer (Malvern Panalytical Ltd., Malvern, Worcestershire, UK) with Cu-Kα radiation (λ = 1.5406 Å) and operating at 40 kV. Diffractogram was obtained at a step size of 0.005° and the samples were scanned over the 2θ range of 5° to 40°.

### 2.8. Thermal Analysis

Differential scanning calorimetry (DSC) and Thermogravimetric analysis (TGA) were conducted using a Discovery 2920 DSC (TA Instruments, New Castle, DE, USA). For DSC, 2 mg of the samples (curcumin, blank PCL scaffold, or curcumin-loaded PCL scaffold) were individually placed in aluminium pans and analysed by DSC with a heating rate of 10 °C/min from 25 to 200 °C under nitrogen atmosphere. For TGA, 5 mg of pure curcumin powder, blank PCL scaffold, or curcumin-loaded PCL scaffold were individually heated from 20 to 500 °C at a heating rate of 10 °C/min under the flow of nitrogen.

### 2.9. Mechanical Properties of Scaffolds

The tensile test was performed on TA. XTplus Texture analyser (Stable Micro Systems, Surrey, UK) with a load cell of 10 kg to identify the tensile properties of the printed scaffolds following ASTM D882 standard. Samples of blank and curcumin-loaded PCL scaffolds (7.0% *w*/*w*) with different thicknesses (0.1 mm, 0.2 mm, 0.4 mm) and pore shapes (honeycomb, square, and triangle) were tested (*n* = 5). The scaffolds were cut into rectangular specimens (10 mm × 60 mm) and the thickness of each specimen was measured at 5 points (Microcaliper, Mitutoyo, Japan). The specimen was gripped at each end for 10 mm and stretched at a strain rate of 5 mm/s until breakage. The force and distance were recorded during the extension of the specimen. The stress-strain curve was automatically generated by the Texture analyser software. The tensile strength was calculated as F/(W·D), where F is force at break, w is width of the specimen at the point of break, and D is specimen thickness. The elongation at break (%) was calculated by E/L·100, where E is the distance of rupture and L is the distance onset of separation. The toughness was calculated by the area underneath the stress-strain curve and the tensile modulus was calculated by the inclination of stress-strain curves in the linear elastic region.

### 2.10. HPLC Analysis of Curcumin

High-performance liquid chromatography (HPLC) was conducted according to a previously developed method [20]. HPLC was performed on a Shimadzu LC system (Shimadzu Corporation, Kyoto, Japan) with DGU-20A3 degasser, SIL-20A HT autosampler, and SPD-M20A DAD detector set at a wavelength of 423 nm. The Waters Symmetry™ C18 column (4.6 × 250 mm, 5 μm particle size) was fitted to the system and the temperature was set to 30 °C. The mobile phase composition was 65%:35% (*v*/*v*) of acetonitrile and citric acid buffer (1% *w*/*v*, pH 3) at a flow rate of 1 mL/min and the injection volume was 20 μL.

### 2.11. Assay of Curcumin Loading in PCL Scaffolds

The drug content of curcumin-loaded scaffolds was determined by HPLC analysis. The scaffolds were cut into pieces (25 × 25 mm) with a weight of 24 ± 1.7 mg (*n* = 3). Each piece was dissolved in 1 mL Tetrahydrofuran (THF) in a sealed container and diluted 100-fold with THF. The dilute solutions (100 μL) were added with 500 μL 65%:35% (*v*/*v*) of acetonitrile and citric acid buffer (1% *w*/*v*, pH 3) and sonicated for 15 min to completely precipitate PCL and dissolve curcumin. The mixtures were then centrifuged, and the supernatants were collected and filtered with 0.22 μm poly (vinylidene difluoride) (PVDF) syringe filters. The filtrates were then analysed by HPLC at a wavelength of 423 nm to measure the concentration of curcumin.

### 2.12. In Vitro Drug Release and Degradation Study

The release profiles of curcumin from the developed scaffolds with different curcumin loadings and geometries were evaluated in vitro for 77 h. Release media was Artificial Cerebrospinal Fluid (ACSF, pH = 7.2) with 0.5% *v*/*v* Tween 80 to increase the solubility of curcumin in ACSF media [30]. The solubility of curcumin in the release media was determined (0.39 mg/mL) to ensure the maintenance of sink conditions. Honeycomb curcumin-loaded scaffolds with different loadings (1.0%, 3.0%, 5.0% and 7.0% w/w, 0.2 mm thickness), 7.0% (*w*/*w*) scaffolds (0.2 mm thickness) with different pore shapes (honeycomb, triangle, square 1 and square 2), and 7.0% (*w*/*w*) honeycomb scaffolds with different thicknesses (0.1, 0.2, and 0.4 mm) were cut into pieces (10 × 10 mm). Each piece (*n* = 4 per group) was weighed, placed into individual amber vials with 20 mL release media, and incubated in a thermostatic incubator (orbital mixer incubator, Ratek Instruments Pty. Ltd., VIC, Australia) at 37 °C with constant horizontal agitation at 175 rpm. The release medium was collected and replaced at various time points (1 h, 4 h, 7 h, 25 h, 49 h, and 77 h). The concentrations of curcumin in the collected release media were determined by HPLC at a wavelength of 423 nm.

The degradation of blank PCL scaffold pieces (10 × 10 mm, *n* = 3) was evaluated by placing them in 20 mL ACSF with 0.5% *v*/*v* Tween 80 and incubated in a thermostatic incubator at 37 °C with constant horizontal agitation at 175 rmp for 8 months; the degradation extents were then observed.

### 2.13. In Vitro Anticancer Activity of Curcumin-Loaded PCL Scaffolds

To evaluate the cytotoxicity of honeycomb curcumin-loaded PCL scaffolds with various loadings, an MTT assay was performed on human primary glioblastoma cell line U87. ACSF media collected from drug release studies at different time points was used for cytotoxicity study. Release media of different loading scaffolds (1%, 3%, 5%, and 7%, *w*/*w*) was collected in 1 h, 4 h, and 24 h and the concentration of curcumin in each sample was determined by HPLC and then used directly for cytotoxicity study. As a positive control, serial solutions of curcumin at different concentration levels (5, 10, 25, 50, and 100 μg/mL) were prepared in release media.

#### 2.13.1. Cell Culture and Maintenance

U87 Cells (Gifted by Professor Stuart Piston, Head of Molecular Signalling Laboratory, the University of South Australia) (passage 9) were cultured in sterile T75 rectangular canted neck cell culture flasks with Minimum Essential Medium Eagle (MEM) media supplemented with 10% FBS, 1% L-glutamine, and 1% sodium pyruvate solution at 37 °C under 5% CO_2_ in a humidified HERACELL 150i CO_2_ incubator (Thermo Fisher scientific, Narellan, NSW, Australia). The media was changed every 48 h. When cells were grown to 80% confluency, the cells were trypsinized and collected after washing with sterile PBS and then passaged at a 1:4 split ratio.

#### 2.13.2. Cytotoxicity Study

MTT cell viability assay was performed to evaluate the cytotoxicity of the curcumin-loaded PCL scaffolds [29,31,32]. Briefly, cells were plated on a 96-well plate at a density of 5 × 10^3^ cells per well in 200 µL culture media and incubated for 24 h under 5% CO_2_ at 37 °C. After 24 h, the tested samples (100 µL) were added into individual wells and incubated for 48 h, along with negative controls including untreated cells (control 1) and cells treated with blank release media (control 2). After 48 h, cells were rinsed gently with PBS, and fresh media was added along with 10 μL of MTT solution (5 mg/mL in sterile PBS, pH 7.40). All plates were incubated for a further 4 h. The MTT solution was then removed from each well and 100 μL of DMSO was added to dissolve any MTT formazan crystals. Plates were covered and left on a plate shaker for 5 min. The absorbance of formazan product was then taken at 570 nm on a Perkin Elmer Wallac plate reader (Perkin Elmer, Inc., Waltham, MA, USA). The results were expressed as percentage viability and groups treated with curcumin were compared to the untreated group (control 1).

## 3. Results and Discussion

### 3.1. FDM Printing of Curcumin-Loaded Scaffolds

Initially, a CAD of the scaffold models was performed. To evaluate the effect of geometry on drug release behaviour and mechanical properties, the scaffolds were designed with a dimension of 60 × 60 mm and with different thicknesses (0.1, 0.2, and 0.4 mm) and pore shape types (honeycomb (Figure 1a), triangle (Figure 1b), square 1 (Figure 1c), and square 2 (Figure 1d)). Goyanes et al. reported that the drug release pattern can be adjusted by altering the surface area to volume (S:V) ratio, which affects the drug diffusion rate from the polymer matrix [33]. Thus, the scaffolds were digitally designed with various geometries to generate models with different surface areas and volumes (Table 3).

Curcumin-loaded filaments were prepared by HME (Figure 2). The blank filament displayed the white colour of PCL, and the curcumin-loaded filaments were of great quality consistency and displayed the colour of curcumin, which varied from yellow to orange with dose increases. With different dosing HME filaments fed to the 3D printer, the PCL scaffolds were FDM 3D printed with high shape fidelity as the CAD models (Figure 3). All scaffolds were printed with a dimension of 60 × 60 mm. The blank and curcumin-loaded scaffolds (1%, 3%, 5%, and 7%, *w*/*w*) were 3D printed with a thickness of 0.2 mm and pore shape honeycomb. For 7.0% (*w*/*w)* curcumin-loaded scaffolds, more diverse geometry designs were printed, with various types of pore shape (honeycomb, triangle, square 1, and square 2, 0.2 mm thickness) and different thicknesses (0.1, 0.2 and 0.4 mm, honeycomb pore shapes). The weight ± SD (*n* = 5) of scaffolds with loadings from 1%, 3%, 5%, and 7% *w*/*w* were 401.2 ± 3.8, 411.4 ± 6.2, 418.2 ± 4.8, and 423.1 ± 2.1 mg, respectively. The diversity of the printed scaffolds demonstrated that 3D printing is a great tool to fabricate versatile drug delivery products with high printing fidelity, providing customised medicine in clinical settings.

### 3.2. Characterisations of 3D Printed Scaffolds

The SEM images of pure curcumin powder, blank, and curcumin-loaded scaffolds (7%, *w*/*w*) are shown in Figure 4. Both blank scaffolds (Figure 4a,b) and curcumin-loaded scaffold (Figure 4c,d) showed micropores on the surfaces, which possibly resulted from the FDM printing process. Figure 4e,f displayed the curcumin powder crystallite. Compared to the blank scaffold (Figure 4a), curcumin-loaded scaffolds (Figure 4c) had a rougher surface. In the SEM topography with 500× magnification (Figure 4d), the curcumin-loaded scaffold showed some particles, which may be drug crystallite that remained on the surface and awaits further investigation by XRD.

To investigate potential interactions of curcumin with PCL matrix, the PA-FTIR spectroscopy was performed on pure curcumin powder, blank, and curcumin-loaded PCL scaffolds (7%, *w*/*w*) (Figure 5). The PCL spectra displayed typical FT-IR peaks at ~2947–2866, 1730, 1471–1367, and 1294 cm^−1^, attributed to stretching vibrations of –CH_2_, stretching vibrations –C=O, bending vibrations of –CH_2_, and stretching vibrations of –COO, respectively [34,35]. Curcumin spectra displayed strong absorbances at 1603 and 1633 cm^−1^ attributed to C=C and –C=O of the aromatic rings [34,36]. The curcumin-loaded scaffold spectra were dominated by PCL peaks with weak curcumin absorptions at 1602 and 1627 cm^−1^. No chemical bonding between curcumin and PCL was observed. Therefore, curcumin was incorporated in polymer PCL compatibly throughout the formulation development process by HME and 3D printing.

The solid-state of curcumin within the PCL matrix was characterised by XRD. As illustrated in Figure 6, the X-ray patterns of the blank scaffold showed typical peaks of PCL at 21.66° and 23.92° [37,38], while pure curcumin exhibited a series of intense and sharp 2θ peaks at 7.90°, 8.98°, 17.20°, 21.22°, 23.32°, and 24.76° [34], indicating that pure curcumin was highly crystalline. Drug-loaded scaffolds did not display any sharp peaks corresponding to curcumin, indicating that curcumin was predominantly dispersed in the PCL matrix amorphously, and the particles on the scaffold surface (Figure 4d) was either drug-polymer dispersion or the amount of drug crystalline was below the sensitivity of detection.

DSC and TGA investigated the distribution of curcumin within the PCL matrix and the effects of the drug on the thermal behaviour of the polymer. As shown in Figure 7, the DSC thermogram for pure curcumin showed an endothermic peak at 184.5 °C corresponding to the melting point of curcumin crystalline [39,40]. The blank and curcumin-loaded PCL scaffolds showed the same melting temperature around 61.5–62.0 °C [41] and no Tm peak of curcumin was observed in the curcumin-loaded PCL scaffold thermogram, suggesting that the majority of the drug was dispersed amorphously in the PCL scaffold and the amount of crystalline curcumin was negligible.

TGA was performed to investigate the thermostability of curcumin and PCL in the scaffolds. TGA thermogram and derivative thermogravimetric (DTG) graphs (Figure 8a,b) displayed a gradual weight loss of curcumin starting from 220 °C and peaked at 295 °C. The weight losses of blank and curcumin-loaded PCL both started from around 325 °C due to thermal degradation of PCL [42] and peaked at 400 °C with approximately half of their initial weight loss. For the curcumin-loaded PCL scaffold, no extra weight loss was observed from 220 °C to 295 °C compared to the blank PCL scaffold. As such, PCL enhanced the thermal stability of curcumin. During the scaffold manufacturing process, elevated temperatures were applied to raw materials, for which HME 3D printing temperature was 95 °C and 3D printing was 130 °C. Therefore, curcumin and PCL were thermally stable during the scaffold manufacturing process via HME and 3D printing, and the PCL matrix offered thermal protection for curcumin.

The effect of loading dose and geometry on the 3D printed scaffold mechanical property was investigated. The stress-strain curves displayed the effect of drug loading and thickness (Figure 9a) and the effect of pore shapes (Figure 9b) on mechanical properties. All curves displayed a unique zig-zag shape, which be attributed to the mesh-structure design. When stress was applied to the scaffold, each structure unit yielded one individually and generated multiple yield points. The mechanical parameters are shown in Table 4. Apart from group square 1, all scaffolds displayed a low tensile modulus of less than 1 Mpa, indicating the softness and flexibility of scaffolds. The 4 to 7-fold elongation at break of all scaffolds indicated their great ductility. Compared to the blank scaffold, the incorporation of curcumin caused a slight decrease in tensile modulus and an increase in elongation strain. The decrease of thickness to 0.1 mm can significantly decrease tensile modulus, indicating the increase of flexibility, while the increase of thickness to 0.4 mm has no significant impact on flexibility, except increasing the elongation. Interestingly, the pore shape has a great impact on scaffold tensile properties. Group square 1 showed a significantly higher tensile modulus of 2 MPa, while the group triangle obtained the lowest elongation strain.

Compared to the stiff polymer matrix of GLIADEL wafer, polyanhydride, which has a high tensile modulus of 3000–5000 Mpa [43], PCL scaffold has a much lower tensile modulus of less than 2 MPa and its flexibility is shown in Figure 10. Thus, 3D printed PCL scaffold obtained great flexibility and stretchability as a soft implant for brain application and the mechanical properties can be easily altered by 3D printing. In the clinic, the scaffold can be printed based on individual tumour size; after resection, the surgeon can tenuously tailor the scaffold to perfectly match the size and shape of the tumour cavity, then attach it to the cavity margin with suturing to eradicate residual tumour cells locally. Therefore, the scaffold has great potential to overcome the mechanical mismatch problem, avoid the risk of detachment, and improve surgical margin coverage [7,12].

### 3.3. Assay of Curcumin Loading in PCL Scaffolds

The drug content in curcumin-loaded scaffolds was determined using HPLC. As shown in Table 5, all the scaffolds achieved percentage recovery of 91–103%, which was within the general USP range of 90–110% of the labelled quantity [44]. Despite different levels of drug loss, efficient drug loading in the scaffold was achieved with the developed method, including solvent casting, HME, and 3D printing. The low standard deviations of all groups indicate that curcumin was uniformly distributed in the PCL matrix.

### 3.4. In Vitro Drug Release and Degradation Study

The release studies of honeycomb scaffolds (1.0%, 3.0%, 5.0%, and 7.0%, *w*/*w*; 0.2 mm thickness) were performed to evaluate the effect of dose on drug release (Figure 11a,b). The release studies of 7.0% (*w*/*w*, 0.2 thickness) scaffold with different pore shapes (honeycomb, triangle, square 1, and square 2), and 7.0% (*w*/*w*, honeycomb) scaffolds with different thicknesses (0.1, 0.2 and 0.4 mm) were performed to investigate the effect of geometry on drug release (Figure 11c,d). All curcumin-loaded scaffolds displayed sustained release pattern over 24 h (thickness 0.2) or 48 h (thickness 0.4). The highest cumulative release was observed in 7% (*w*/*w*, 0.4 thickness) honeycomb over 77 h (1346 µg), which was attributed to the highest amount of curcumin present in those scaffolds. This pattern of a large amount of curcumin release in 48 h is beneficial for the elimination of residual cancer cells because of the dose-dependent anti-glioma effects of curcumin and its low toxicity profiles, which allows it to be safely used in high dose [25]. The release data of each group was fitted in Monoexponential, Higuchi, Niebergall, Hixson-crowel, Weibull, and Biexponential models and the release profiles were all consistent with the biexponential release mechanism (Table 6).

#### 3.4.1. The Effect of Drug Loading on Drug Release

The release studies of curcumin-loaded PCL scaffolds (honeycomb, thickness 0.2 mm) with different drug loadings (1, 3, 5 and 7%) was performed to evaluate the effect of drug loading on release behaviour. Despite different loadings, all curcumin-loaded PCL scaffolds displayed a sustained release of approximately 75% of the drug over 24 h and drug release was completed in 48 h. As the drug loading increased, the release amount increased proportionally while the release curvilinear trend remained unchanged. By altering the drug loading dose, the drug release profile can be controlled to fit in the therapeutic windows of individual patients. Therefore, the 3D printed PCL scaffold is a great carrier to deliver a personalised dose according to individual regimens.

#### 3.4.2. The Effect of Geometry on Drug Release

The release studies of curcumin-loaded PCL scaffolds (7%, *w*/*w*) with different thicknesses and pore shapes were performed to evaluate the effect of geometry on drug release (Figure 11c,d). As the thickness increased from 0.1, 0.2 to 0.4 mm, the sustained release durations were significantly prolonged from 6 h to, 24, to 48 h, which can account for the decrease of the S:V ratio [33]. In addition, as thickness increased, the amount of drug released increased from 245 µg to 599 µg and 1346 µg.

Compared to altering the thickness, changing the pore shapes has a minor effect on drug release durations but a bigger effect on drug release amounts. While thickness remained the same, Group square 1 and triangle released higher levels of curcumin compared to group square 2 and honeycomb scaffolds, which was attributed to the highest amount of curcumin present in each 10 × 10 mm release samples. The 60 × 60 mm CAD models of group square 1 and triangle have bigger volumes (Table 3), hence more drug was present in 10 × 10 mm samples, leading to more drug release. In addition to adjusting the dose by 3D printing with different loading filaments, here dose adjustment can also be achieved by CAD and 3D printing alone, by altering the scaffold volume per unite area or increasing the thickness to both increase drug release amount and duration. Therefore, 3D printing is a great tool for the versatile manufacturing of scaffolds to adjust the treatment regimen according to individual needs.

#### 3.4.3. In Vitro Degradation Study

In vitro degradation study was performed with blank PCL scaffold pieces incubated in the release media in a thermostatic incubator at 37 °C with constant agitation at 175 rpm for 8 months. During the observation period, the scaffolds became fragile after 6 months of incubation (Figure 12b); after 8 months of incubation, it was observed that most parts of the resulting scaffolds had disintegrated into flocculent masses (Figure 12c,d). According to previous reports, PCL has a slow biodegradation rate and the full degradation time of PCL varies from several months to several years, depending on the size, porosity, molecular weight, and degradation condition [45,46], Therefore, the extent of the scaffold degradation in the present in vitro study was as expected for an eight-month incubation period. Faster degradation is expected in vivo after surgical implantation in the brain due to physical sink conditions and enzymatic hydrolysis [19].

### 3.5. In Vitro Anticancer Activity of Curcumin-Loaded PCL Scaffolds

To evaluate the anticancer activity of curcumin released from the 3D printed PCL scaffolds with different loadings (1%, 3%, 5%, 7%, *w*/*w*), human GBM cell line U87 cells were utilised. First, the cytotoxicity of standard curcumin serial solutions was assessed by treating the cells with various concentrations from 5 to 100 µg/mL for 48 h and cell viability was characterised by MTT assay. The results (Figure 13) displayed a concentration-dependent inhibition on cell viability and even a low concentration of curcumin (5 µg/mL) showed significant cytotoxicity with 53.4% viability. Thus, curcumin is a promising agent against U87 cells.

The cytotoxicity of curcumin released from the 3D printed PCL scaffolds with different loadings (1%, 3%, 5%, 7%, *w*/*w*) against U87 cells for 48 h treatment was subsequently evaluated by MTT assay (Figure 14a). Untreated cells (control 1) and cells treated with blank release media (control 2) were also included for comparison. Besides, the cytotoxic effect of different concentrations of curcumin released from different loadings of PCL scaffold for various release durations was compared to the cytotoxicity of their respective curcumin concentrations (positive control) (Figure 14b).

The cell viability of the group treated with blank release media (control 2) was 96.5% compared to the untreated group (control 1), indicating that the possible cytotoxicity of release media can be excluded. Curcumin has significantly inhibited the proliferation of U87 cells following 48 h of exposure. Curcumin has a concentration-dependent effect on cell survival of U87 cells from a concentration as low as 5 µg/mL. With release time increased from 1 h to 24 h or dose increased from 1% to 7%, cell viability showed an obvious decreasing trend (Figure 14) and there were statistically significant differences observed in all treated groups compared to the untreated control groups. With a higher dose or release time, U87 cells had higher concentration (Table 7), and greater inhibition. Compared to standard curcumin serial solutions with respective concentrations, for most concentration levels, no statistical significance was observed between the viability results of scaffold-released curcumin and pure curcumin. However, overall, the scaffold-released curcumin displayed a greater inhibition effect against U87, and an interesting phenomenon was observed on concentration level 100 µg/mL, where scaffold-released curcumin showed significantly higher cytotoxicity compared to pure curcumin. Therefore, the curcumin-loaded scaffolds demonstrated great cytotoxic effects against U87 cells. The scaffolds can maintain cytotoxic effects throughout at least 24 h release duration and the released curcumin can retain its biological activity for 48 h in vitro, which is promising for the application of GBM recurrence prevention after surgery.

## 4. Conclusions

A novel biodegradable soft scaffold was developed using extrusion-based 3D printing technology for the localised administration of curcumin after resection surgery for GBM treatment. After the solvent casting of curcumin-loaded PCL film and extrusion of curcumin/PCL filament, curcumin-loaded PCL porous scaffolds were FDM 3D printed with different loading doses and geometries, obtaining versatile spatiotemporal characteristics. The polymeric scaffolds developed by FDM 3D printing technology achieved great flexibility and stretchability, which can avoid the mechanical mismatch problem of the GLIADEL wafer and further improve the coverage of irregular surgical margin. The scaffolds developed in the present study achieved varied release patterns of curcumin with adjustable release amounts and durations, demonstrating the versatility and dirigibility of 3D printing technology. An in vitro anticancer cell study has demonstrated the significant cytotoxic effect of curcumin-loaded scaffolds against the U87 human GBM cell line. Therefore, the 3D printed scaffolds for curcumin local administration have great potential to achieve personalised treatment of GBM, prevent post-surgery GBM recurrence, and eventually prolong the life expectancy of GBM patients.

## Figures and Tables

**Figure 1 pharmaceutics-13-00471-f001:**
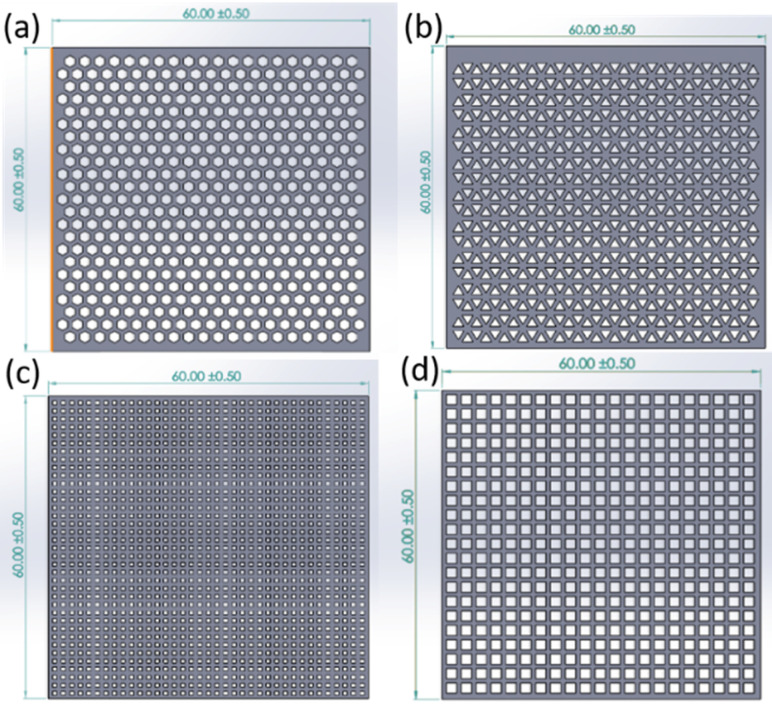
Computer-aided design (CAD) of curcumin-loaded scaffold (60 × 60 mm) with different pore types: (**a**) honeycomb, inner circle diameter = 2.0 mm; (**b**) triangle, inner circle diameter = 1.2 mm; (**c**) square 1, inner circle diameter = 0.8 mm and (**d**) square 2, inner circle diameter = 2.0 mm.

**Figure 2 pharmaceutics-13-00471-f002:**
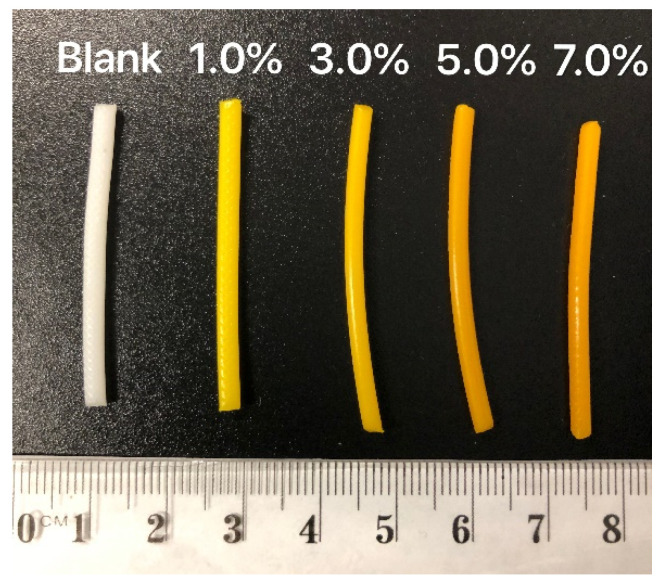
Blank and curcumin-loaded PCL filaments (1%, 3%, 5%, and 7%, *w*/*w*).

**Figure 3 pharmaceutics-13-00471-f003:**
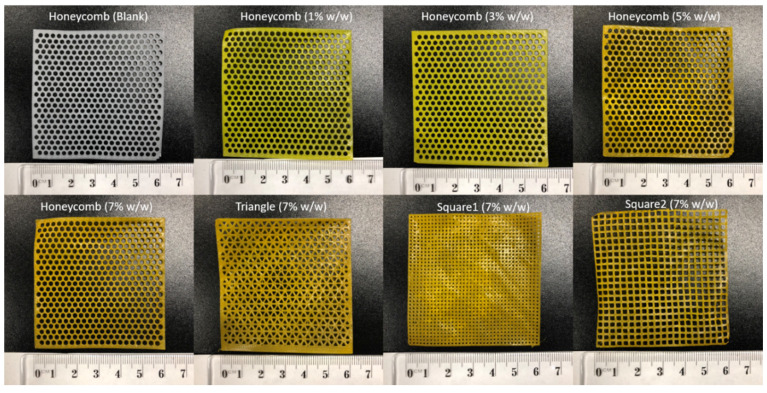
3D printed blank and curcumin-loaded scaffolds (0.2 mm thickness) with different drug loadings (1%, 3%, 5%, and 7%, *w*/*w*) and pore shape types (honeycomb, triangle, square 1, and square 2).

**Figure 4 pharmaceutics-13-00471-f004:**
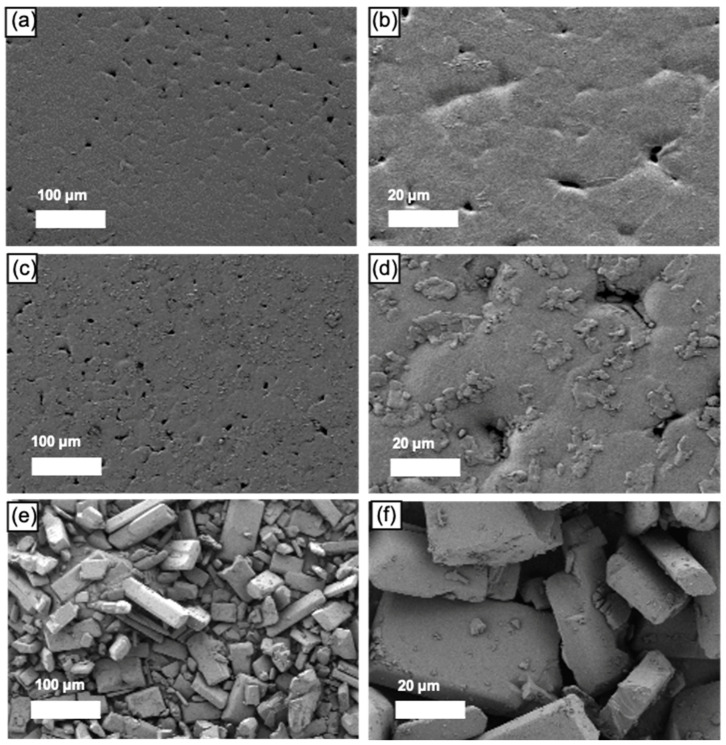
Scanning Electron Microscopy (SEM) images of (**a**) blank scaffold with 100× magnification and (**b**) with 500× magnification; (**c**) curcumin-loaded PCL scaffold with 100× magnification and (**d**) with 500× magnification; (**e**) pure curcumin powder with 100× magnification and (**f**) with 500× magnification.

**Figure 5 pharmaceutics-13-00471-f005:**
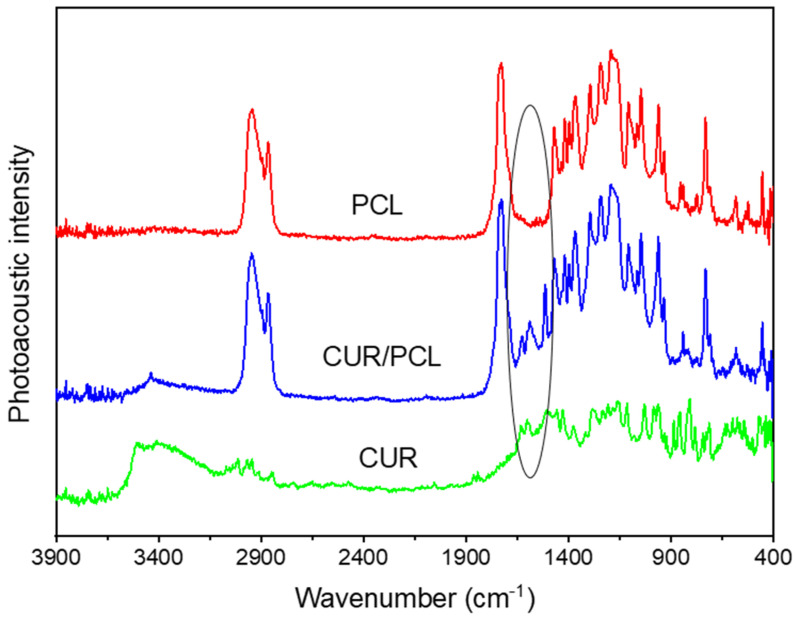
Photoacoustic Fourier-transform infrared (PA-FTIR) spectra of pure curcumin (CUR), blank PCL scaffold, and curcumin-loaded PCL scaffold (7%, *w*/*w*).

**Figure 6 pharmaceutics-13-00471-f006:**
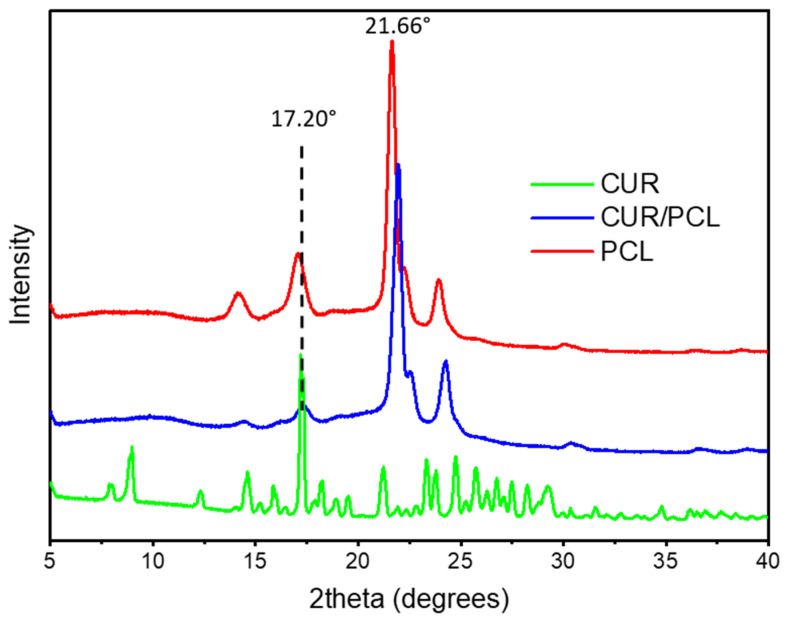
X-ray diffractograms of pure curcumin (CUR), blank PCL scaffold, and curcumin-loaded PCL scaffold (7%, *w*/*w*).

**Figure 7 pharmaceutics-13-00471-f007:**
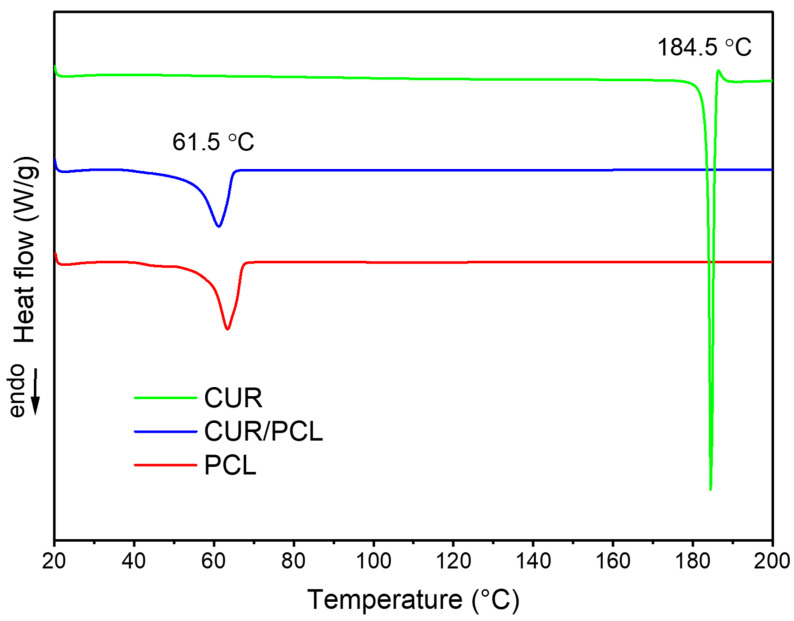
Differential scanning calorimetry (DSC) thermograms of pure curcumin (CUR), blank, and curcumin-loaded PCL scaffolds (7%, *w*/*w*).

**Figure 8 pharmaceutics-13-00471-f008:**
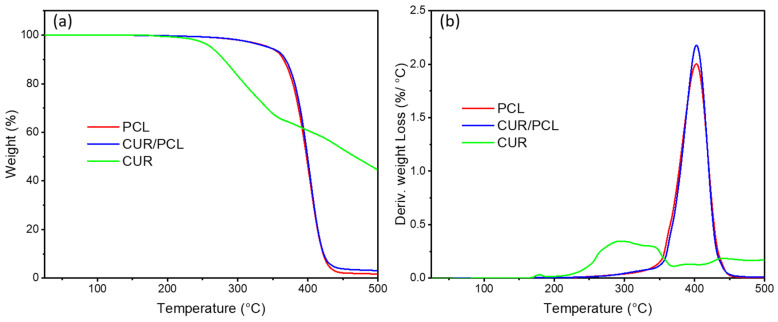
(**a**) Thermogravimetric analysis (TGA) thermograms and (**b**) derivative thermogravimetric graphs of curcumin (CUR), blank, and curcumin-loaded PCL scaffold (7%, *w*/*w*).

**Figure 9 pharmaceutics-13-00471-f009:**
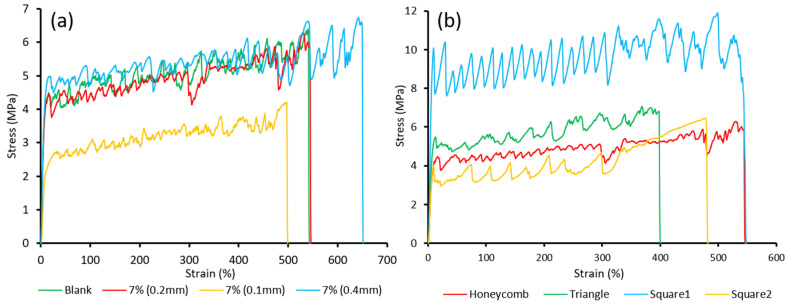
Stress-strain curves of 3D printed scaffolds showing (**a**) the effect of dose and thickness (**b**) the effect of pore shapes.

**Figure 10 pharmaceutics-13-00471-f010:**
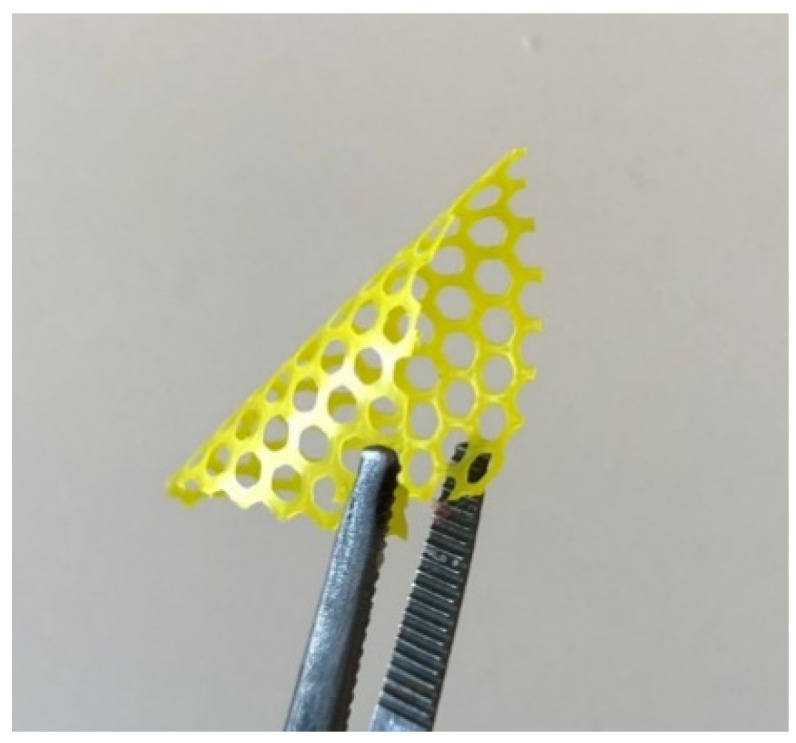
The flexible property of 1% (*w*/*w*) honeycomb scaffold.

**Figure 11 pharmaceutics-13-00471-f011:**
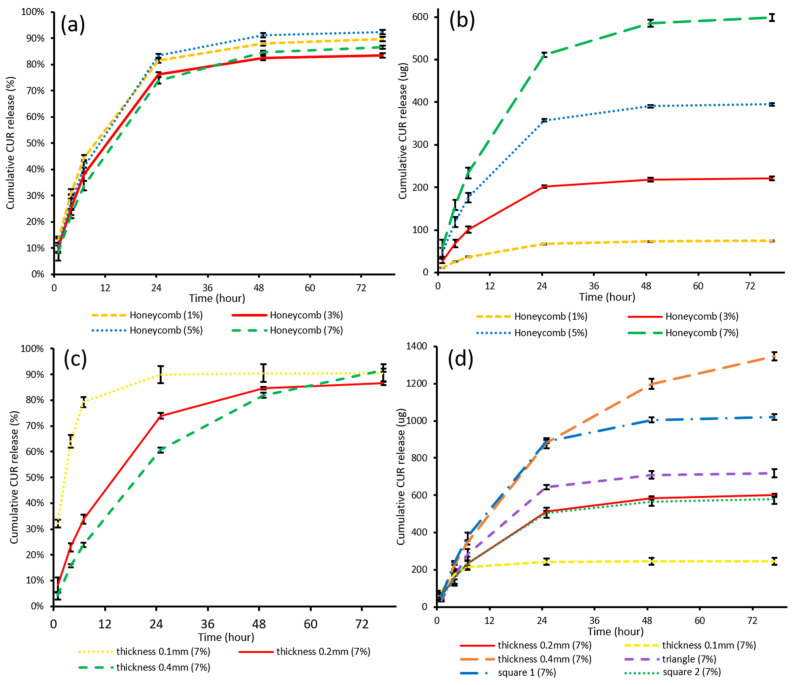
In vitro release profiles of curcumin-loaded PCL scaffolds: (**a**) the cumulative release percentages of scaffolds with loadings 1%, 3%, 3%, and 7%, *w*/*w*; (**b**) the cumulative release amount of scaffolds with loadings 1%, 3%, 3%, and 7% *w*/*w*; (**c**) the cumulative release percentages of 7% (*w*/*w*) scaffolds with thicknesses 0.1, 0.2, and 0.4 mm; (**d**) the cumulative release amount of 7% (*w*/*w*) scaffolds with thicknesses 0.1, 0.2, 0.4 mm and pore shapes honeycomb, triangle, square 1 and square 2.

**Figure 12 pharmaceutics-13-00471-f012:**
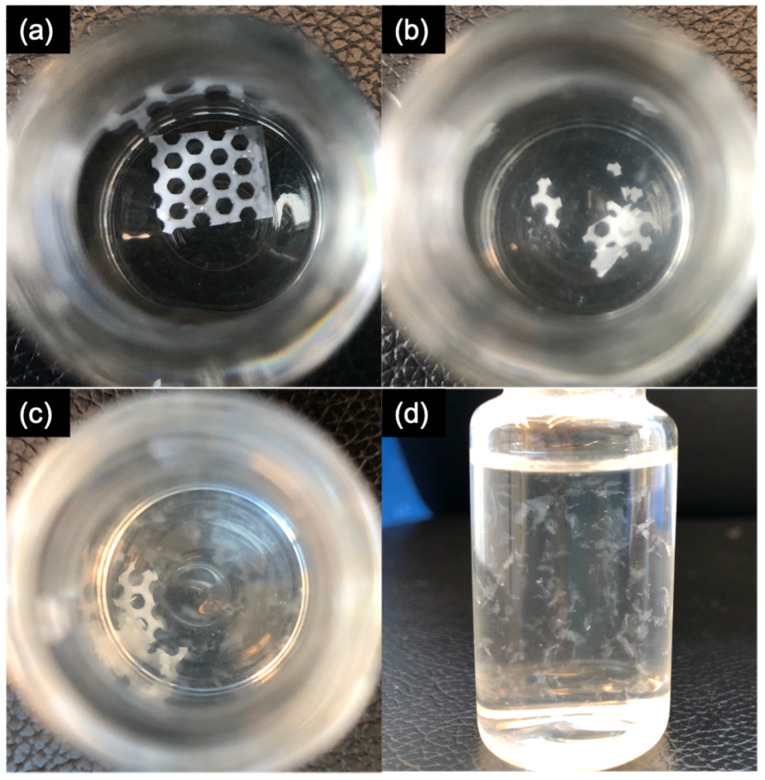
(**a**) Top view of blank PCL scaffold at the start of incubation; (**b**) top view of blank PCL scaffold after six-month incubation; (**c**) top-view of blank PCL scaffold after eight-month incubation; (**d**) side-view of blank PCL scaffold after eight-month incubation.

**Figure 13 pharmaceutics-13-00471-f013:**
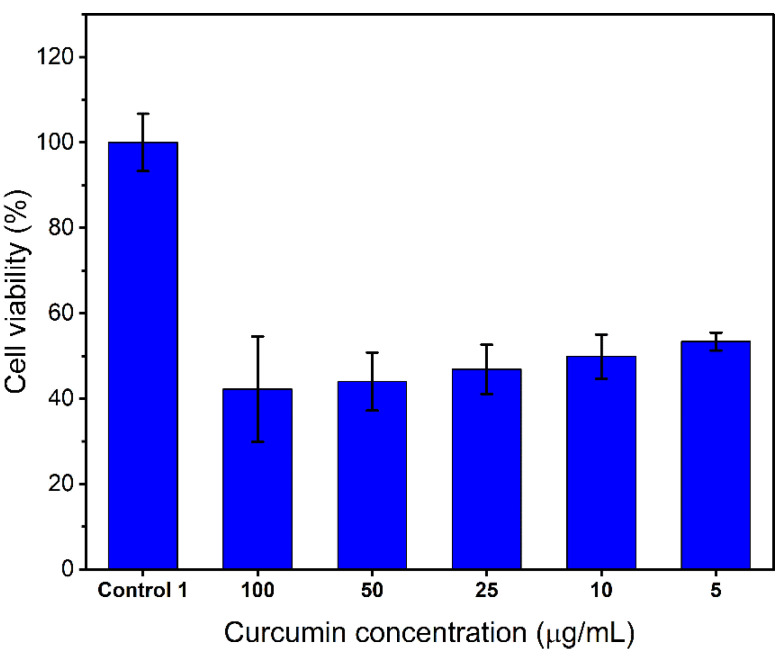
Cytotoxic effects of various concentrations of curcumin in serial solutions (positive control) on the U87 cell line after 48-h treatment. Results from three replicate measurements are presented as mean ± SD.

**Figure 14 pharmaceutics-13-00471-f014:**
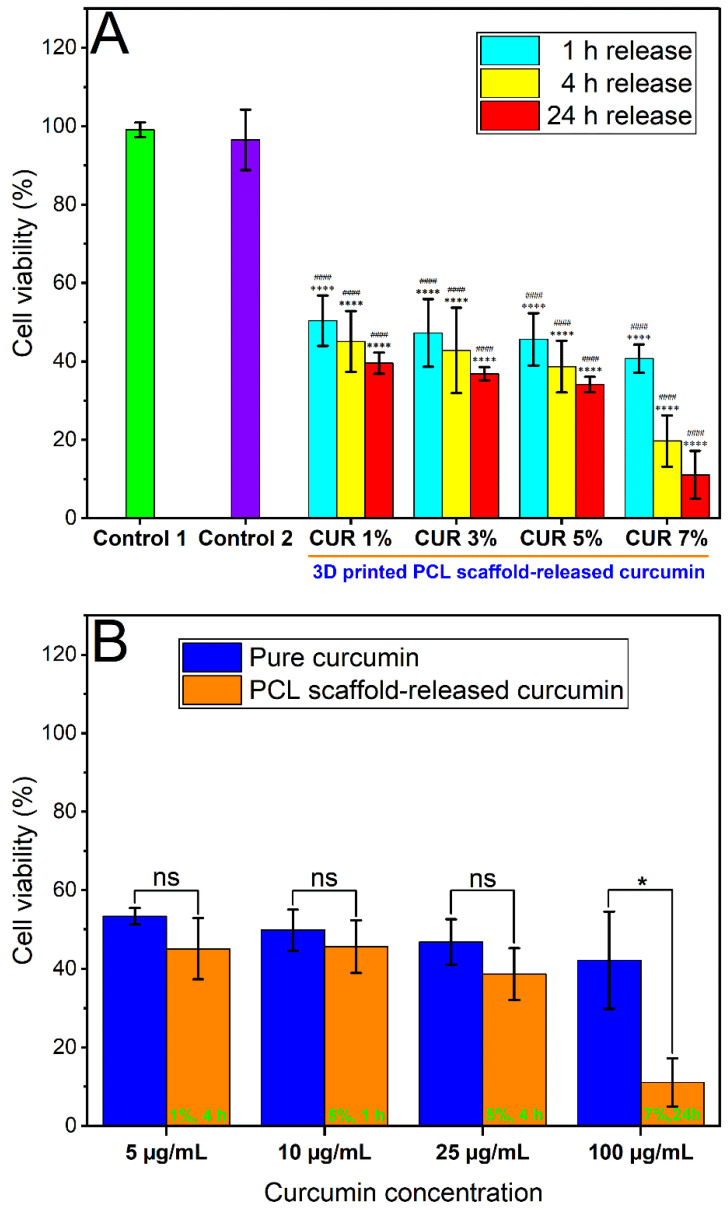
(**a**) Cytotoxic effects of curcumin in release media acquired in different durations (1 h, 4 h, 24 h) from 3D printed scaffolds with different loadings (1%, 3%, 5%, 7%, *w*/*w*). (**b**) Comparison of cytotoxic effects of different concentrations of curcumin released from different loadings and release durations of PCL scaffold, with the respective curcumin concentrations (positive control) on the U87 cell line after 48 h treatment. Results from at least three replicate measurements are presented as mean ± SD. *p*-values were obtained via one-way analysis of variance (ANOVA). ns *p* > 0.05, * *p* ≤ 0.05. **** *p* < 0.0001 and #### *p* < 0.0001 compared to the untreated group (control 1) and group treated with blank release media (control 2), respectively.

**Table 1 pharmaceutics-13-00471-t001:** Amount of curcumin, poly(caprolactone) (PCL), and tetrahydrofuran (THF) used for solvent casting of curcumin-loaded PCL film.

Loading Rate	Curcumin (g)	PCL (g)	THF (mL)
1%	0.5299	49.535	250
3%	1.5106	48.463	250
5%	2.5254	47.512	250
7%	7.0945	92.150	500

**Table 2 pharmaceutics-13-00471-t002:** Key parameters of 3D printing.

Parameters	Value
Printing temperature	130 °C
Build plate temperature	45 °C
Printing speed	20 mm/s
Flow	120%
Layer height	0.1 mm

**Table 3 pharmaceutics-13-00471-t003:** Surface area, volume, and Surface to Volume (S:V) ratio of digital scaffold models with different pore shapes (honeycomb, triangle, square 1, and square 2) and thicknesses (0.1, 0.2, and 0.4 mm).

Parameter	Honeycomb (0.1 mm)	Honeycomb (0.2 mm)	Honeycomb (0.4 mm)	Triangle (0.2 mm)	Square 1 (0.2 mm)	Square 2 (0.2 mm)
Surface area (mm^2^)	4287.13	4637.45	5338.09	5766.47	6371.84	4425.6
Volume (mm^3^)	196.84	393.68	787.36	497.77	544.77	367.2
S:V ratio	21.78	11.78	6.78	11.58	11.70	12.05

**Table 4 pharmaceutics-13-00471-t004:** Mechanical parameters of the blank honeycomb scaffold, 7% (*w*/*w*) honeycomb scaffolds with thickness 0.1, 0.2, and 0.4 mm; 7% (*w*/*w*) triangle, square 1, and square 2 scaffolds with thickness 0.2 mm (*n* = 5 per group).

Scaffold	Thickness (mm)	Tensile Strength (Mpa)	Elongation at Break (%)	Toughness (MJ/m^3^)	Tensile Modulus (Mpa)
Blank honeycomb	0.2	6.72 ± 0.72	586.65 ± 30.65	28.02 ± 3.89	0.87 ± 0.16
7% honeycomb	0.2	6.80 ± 0.49	657.96 ± 38.11	30.45 ± 3.62	0.78 ± 0.08
7% honeycomb	0.4	6.60 ± 0.16	700.28 ± 32.55	33.64 ± 1.56	0.80 ± 0.04
7% honeycomb	0.1	5.09 ± 0.99	611.90 ± 38.43	19.95 ± 4.43	0.44 ± 0.12
7% Triangle	0.2	7.18 ± 0.60	437.86 ± 63.05	21.71 ± 4.18	0.96 ± 0.03
7% Square 1	0.2	12.28 ± 0.57	660.57 ± 92.09	57.20 ± 8.88	1.79 ± 0.08
7% Square 2	0.2	6.44 ± 0.59	641.55 ± 93.34	24.53 ± 4.20	0.72 ± 0.19

**Table 5 pharmaceutics-13-00471-t005:** Theoretical and practical amounts, percentage recovery of curcumin in different loading scaffolds (*n* = 3).

Theoretical Drug Loading	Theoretical Amount of Curcumin (mg) ± SD	Percentage Recovery (%)	Practical Drug Loading
1.0%	0.22 ± 0.01	97.4 ± 3.2	1.0%
2.9%	0.60 ± 0.04	91.1 ± 2.4	2.6%
5.0%	1.27 ± 0.01	91.6 ± 1.4	4.5%
6.8%	1.67 ± 0.07	102.9 ± 1.3	7.0%

**Table 6 pharmaceutics-13-00471-t006:** Equation and correlation coefficient (R^2^) of scaffolds with different curcumin loadings (1%, 3%, 5%, 7%, *w*/*w*, honeycomb), 7% (*w*/*w*) scaffolds with different thickness (0.1, 0.2, 0.4 mm, honeycomb), and 7% (*w*/*w*) scaffolds with different pore shapes (honeycomb, triangle, square 1, square 2, thickness = 0.2 mm) of curcumin release profile in release kinetic models Monoexponential, Higuchi, Niebergull, Hixcon-crowel, Weibull and biexponential models.

Group	Parameter	Biexponential	Monoexponential	Higuchi	Niebergull	Hixcon-Crowell	Weibull
**1%**	**Equation**	1 − Q = 0.8392e^−0.09022t^ + 0.1029e^−0.00013t^	ln(1 − Q) = −0.0122t − 0.7801	Q = 0.0616t^1/2^ + 0.272	(1 − Q)^1/2^ = −0.0032t + 0.7128	(1 − Q)^1/3^ = −0.0026t + 0.7896	lnln(1/1 − Q) = 0.5677 lnt − 1.7174
	**R^2^**	0.9997	0.6348	0.7450	0.5828	0.6001	0.9384
**3%**	**Equation**	1 − Q = 0.8132e^−0.08696t^ + 0.1623e^−0.00013t^	ln(1 − Q) = −0.0094t − 0.6488	Q = 0.0600t^1/2^ + 0.2265	(1 − Q)^1/2^ = −0.0028t + 0.7512	(1 − Q)^1/3^ = −0.0022t + 0.8198	lnln(1/1 − Q) = 0.5724 lnt − 1.9534
	**R^2^**	0.9987	0.5972	0.7416	0.5631	0.5745	0.926
**5%**	**Equations**	1 − Q = 0.9019e^−0.08392t^ + 0.07276e^−0.00013t^	ln(1 − Q) = −0.0144t − 0.7972	Q = 0.0672t^1/2^ + 0.2408	(1 − Q)^1/2^ = −0.0036t + 0.7170	(1-Q)^1/3^ = −0.0030t + 0.7906	lnln(1/1 − Q) = 0.6431 lnt − 1.9379
	**R^2^**	0.9988	0.6452	0.7488	0.589	0.6076	0.9409
**7%**	**Equations**	1 − Q = 0.8558e^−0.07196t^ + 0.1289e^−0.00013t^	ln(1 − Q) = −0.0115t − 0.6068	Q = 0.0667t^1/2^ + 0.1853	(1 − Q)^1/2^ = −0.0033t + 0.7671	(1 − Q)^1/3^ = −0.0026t + 0.8314	lnln(1/1 − Q) = 0.6481 lnt − 2.2056
	**R^2^**	0.9992	0.6672	0.7797	0.6183	0.6347	0.9457
**0.1 mm**	**Equations**	1 − Q = 0.7630e^−0.2701t^ + 0.0969e^−0.00013t^	ln(1 − Q) = −0.0083t − 1.3663	Q = 0.0344t^1/2^ + 0.5676	(1 − Q)^1/2^ = −0.0019t + 0.5389	(1 − Q)^1/3^ = −0.0017t + 0.6529	lnln(1/1 − Q) = 0.3225 lnt − 0.5232
	**R^2^**	0.9997	0.4436	0.5211	0.3911	0.4095	0.8181
**0.4 mm**	**Equations**	1 − Q = 0.8594e^−0.04373t^ + 0.1339e^−0.01136t^	ln(1 − Q) = −0.0235t − 0.2619	Q = 0.084t^1/2^ + 0.0615	(1 − Q)^1/2^ = −0.0051t + 0.8425	(1 − Q)^1/3^ = −0.0044t + 0.8952	lnln(1/1 − Q) = 0.8916 lnt − 3.0397
	**R^2^**	0.9998	0.9605	0.8985	0.8457	0.8884	0.9954
**Triangle**	**Equations**	1 − Q = 1.0588e^−0.08433t^ + 0.0080e^−0.00014t^	ln(1 − Q) = −0.0631t − 0.8518	Q = 0.0793t^1/2^ + 0.1954	(1 − Q)^1/2^ = −0.0056t + 0.7015	(1 − Q)^1/3^ = −0.0058t + 0.7715	lnln(1/1 − Q) = 1.0786 lnt − 2.836
	**R^2^**	0.9972	0.8379	0.7467	0.6588	0.7269	0.9683
**Square1**	**Equations**	1 − Q = 1.0377e^−0.07842t^ + 0.00034e^−0.00012t^	ln(1 − Q) = −0.0632t − 0.8478	Q = 0.0794t^1/2^ + 0.192	(1 − Q)^1/2^ = −0.0056t + 0.7067	(1 − Q)^1/3^ = −0.0058t + 0.7292	lnln(1/1 − Q) = 1.051 lnt − 2.7318
	**R^2^**	0.998	0.8355	0.7593	0.6666	0.7292	0.984
**Square2**	**Equations**	1 − Q = 0.9167e^−0.07795t^ + 0.2587e^−0.00012t^	ln(1 − Q) = −0.0145t − 0.7127	Q = 0.07t^1/2^ + 0.2044	(1 − Q)^1/2^ = −0.0037t + 0.7402	(1 − Q)^1/3^ = −0.0031t + 0.8091	lnln(1/1 − Q) = 0.6912 lnt − 2.1859
	**R^2^**	0.9993	0.6779	0.7664	0.6145	0.6355	0.9452

* Q: Cumulative release percentage of curcumin from PCL scaffold; t: time (hours).

**Table 7 pharmaceutics-13-00471-t007:** The curcumin equivalent concentrations to the cell culture media of release samples acquired in different durations (1 h, 4 h, 24 h) from 3D printed scaffolds with different loadings (1%, 3%, 5%, 7%, *w*/*w)* analysed via HPLC.

Time	1 h	4 h	24 h
Theoretical Curcumin Loading	Concentration (µg/mL)		
**1%**	1.56 ± 0.05	5.87 ± 0.12	12.29 ± 0.85
**3%**	3.25 ± 0.07	17.68 ± 0.11	35.46 ± 1.38
**5%**	8.93 ± 0.18	26.04 ± 0.49	69.78 ± 1.41
**7%**	12.17 ± 0.43	39.43 ± 1.47	101.88 ± 4.86

## Data Availability

The authors confirm that the data supporting the findings of this study are available within the article.

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
