# Peer review of "Three-Dimensional Printing of Curcumin-Loaded Biodegradable and Flexible Scaffold for Intracranial Therapy of Glioblastoma Multiforme"

_pharmaceutics, 2021, doi:10.3390/pharmaceutics13040471_

Round 1

Reviewer 1 Report

The authors have manufactured PCL scaffolds by 3D printing and incorporated curcumin as an anticancer agent. They have characterized the resulting composites in detail, including the in vitro biodegradability, and evaluated the durg effect and viability with target cancer cells. Their results and discussion are well presented and clear, and show interesting potential for anticancer applications. I strongly suggest the publication of the articles as it is after few minor revisions and comments.

  1. The tensile assays should be repeated, the plots do not show the typical elastic curve. There is too much noise that should be avoided.
  2. Please avoid the use of a lot of acronyms.
  3. What does “percentage of recovery” state for in section 3.3? Which deformation/process did the material suffer? Please clarify.
  4. Table 6 should be expanded into a full wide page range to facilitate its understanding. Also specify the meaning of “t”.

Author Response

1 The tensile assays should be repeated, the plots do not show the typical elastic curve. There is too much noise that should be avoided.

Thank you for your valuable question. The tensile results displayed a unique zig-zag shape curve due to the mesh-structure of the scaffold. When the scaffold was stretched during the tensile assay, each structure unit yielded individually and one after another and displayed this “noise” like curve with multiple yield points. This part has been further explained and revised accordingly (Line 355)

2 Please avoid the use of a lot of acronyms.

We have followed the comment and revised. The mostly used acronym “CUR” has been changed to “curcumin”. PCL was unchanged because it’s a commonly used acronym.

3 What does “percentage of recovery” state for in section 3.3? Which deformation/process did the material suffer? Please clarify.

Here percentage of recovery = amount of curcumin recovered from the scaffold sample/the theoretical amount of curcumin in the scaffold sample. The theoretical amount of curcumin= sample weight × (curcumin weight added/total weight of curcumin and PCL). The weights of curcumin and PCL during the drug development process are displayed in Table 1. The sample didn’t go through deformation, instead it was dissolved in THF solvent and mobile phase was added to precipitate polymer and dissolve drug. The extracted drug was measured by HPLC and the extracted drug was calculated as percentage of recovery. More detail is in section 2.11.

4 Table 6 should be expanded into a full wide page range to facilitate its understanding. Also specify the meaning of “t”.

We have followed the comment and revised (Line 472)

Reviewer 2 Report

The authors have presented a new treatment option for GBM using 3D printed PCL mesh loaded with curcumin.

The manuscript it well presented and experiments are well organized. How ever, few improvements are needed.

In the introduction, please include other studies that used PCL and similar biomaterials for 3D printed drugs drug delivery systems such as- Farmer et al 2020, Ballard et al 2018, Tappa et al 2017, etc.

Fig 2- Please add scale bar.

Author Response

1 In the introduction, please include other studies that used PCL and similar biomaterials for 3D printed drugs drug delivery systems such as- Farmer et al 2020, Ballard et al 2018, Tappa et al 2017, etc.

Thank you for your valuable suggestion. We have followed the comment and revised (Line 64).

2 Fig 2- Please add scale bar.

We have followed the comment and revised (Line 280).

Reviewer 3 Report

Overall, the manuscript is well written and the scientific content of interest to the readers of pahrmaceutics. <however, there are few concerns that need revision before publication. 

  1. Regarding the methodology, the section about the texture analyser is barely explained. Please, add more details about how you perform the measurements. Figure 9 shows a very poor baseline. Did you use a three point bed adaptor? How did you calculate all the parameters shown in table 4? How did you ensure the filaments and nets did not move or slide how did you keep them stable?
  2. Also, you have to explain a bit more about the rational of using PCL as excipient, it does not make sense that the release of the drug occurs in 72 h but the polymer stays for many months which can cause inflammation. 
  3. Also, you have to explain the rational why do you need to preform the films by solvent casting adn then extrude them. Is that the content homogenity was not good if you extrude the powder mixture directly? Please explain. 
  4. You need to do statistics in your results. 
  5. You need to add a discussion, how your results compare to those obtained by other authors. Have you considered to add the CUR directly by passive diffusion in your filaments? similar drug loadings can be achieved, have a look to this paper: Personalised 3D printed medicines: Optimising material properties for successful passive diffusion loading of filaments for fused deposition modelling of solid dosage forms. 

Author Response

1 Regarding the methodology, the section about the texture analyser is barely explained. Please, add more details about how you perform the measurements. Figure 9 shows a very poor baseline. Did you use a three point bed adaptor? How did you calculate all the parameters shown in table 4? How did you ensure the filaments and nets did not move or slide how did you keep them stable?

Thank you for your valuable questions. We have followed the comment and revised the methodology. The tensile results displayed a unique zig-zag shape curve due to the mesh-structure of the scaffold. When the scaffold was stretched during the tensile assay, each mesh-structure unit yielded individually and one after another thus displayed this “noise” like curve with multiple yield points. The tensile grips are two vertical grips and the upper grip moved up during the tensile test. For the scaffold, the scaffold strip was tightly held at each end by the tensile grips, so it didn’t move except elongate vertically. No tensile assay was performed on filament. The Parameters were calculated automatically via the Texture analyser software where there are set program based on the ASTM D882 standard. To further clarify the method, parameter calculation methods have been added in the methodology part.

2 Also, you have to explain a bit more about the rational of using PCL as excipient, it does not make sense that the release of the drug occurs in 72 h but the polymer stays for many months which can cause inflammation. 

Thank you for your suggestion. PCL is an FDA-proved biodegradable polymer for human implantable use, therefore it’s safety use has been guaranteed. Among all FDA-proved biodegradable polymers, PCL has the slowest degradation rate but also the greatest flexibility. We chose PCL as the matrix of biodegradable flexible scaffold because PCL is a compromise between flexibility and biodegradability. Due to high mortality rate of GBM, it is crucial for patients to receive a treatment with high antitumor efficiency and low toxicity within 1-2 months, so polymer stays might be less of an issue. However, in future project we will explore the possibility of co-polymer to decrease the degradation time. We have followed the comment and revised this part (Line 63).

3 Also, you have to explain the rational why do you need to preform the films by solvent casting adn then extrude them. Is that the content homogenity was not good if you extrude the powder mixture directly? Please explain. 

The preparation of solvent cast film before extrusion is to mix drug and polymer first to improve homogeneity. In preliminary study, direct extrusion of drug and polymer mixture generate filaments with bad uniformity. This part has been further explained and revised accordingly (Line 111).

4 You need to do statistics in your results. 

Thank you for your valuable suggestion. For Table 3 the value is automatically generated by the model great software thus there is no statistics and it has been further explained and revised accordingly (Line 135).For Table 7 the statistics were added according to this suggestion, presented as mean±sd (Line 526).

5 You need to add a discussion, how your results compare to those obtained by other authors. Have you considered to add the CUR directly by passive diffusion in your filaments? similar drug loadings can be achieved, have a look to this paper: Personalised 3D printed medicines: Optimising material properties for successful passive diffusion loading of filaments for fused deposition modelling of solid dosage forms. 

Thank you for giving this example paper. We do make a comparison between GLIADEL wafer and our formulation. Our scaffold demonstrated similar drug release range (<1 week) to the wafer but much better mechanical performance. Also, 3D printing brings the advantage of personalisation. We have considered passive diffusion as a drug loading method. however the loading for passive diffusion is limited (1-2% as suggested in this paper) and it might be a challenge to find the right solvent. Therefore, we choose direct filament extrusion.